



Atmospheric
Chemistry
and Physics

# Clear-air lidar dark band

**Paolo Di Girolamo**[1], **Andrea Scoccione**[2], **Marco Cacciani**[2], **Donato Summa**[1], **Benedetto De Rosa**[1], and
**Jan H. Schween**[TS1][3]

[1]Scuola di Ingegneria, Università degli Studi della Basilicata, Potenza, Italy
[2]Dipartimento di Fisica, Università di Roma "La Sapienza", Rome, Italy
[3]Institut fuer Geophysik und Meteorologie, Universität zu Köln, Cologne, Germany

**Correspondence:** Paolo Di Girolamo (digirolamo@unibas.it)

**Abstract.** This paper illustrates measurements carried out by the Raman lidar BASIL in the frame of the HD(CP)$^2$ Observational Prototype Experiment (HOPE), revealing the presence of a clear-air dark band phenomenon (i.e. a minimum in lidar backscatter echoes) in the upper portion of the convective boundary layer. The phenomenon is clearly distinguishable in the lidar backscatter echoes at 532 and 1064 nm, as well as in the particle depolarisation data. This phenomenon is attributed to the presence of lignite aerosol particles advected from the surrounding open pit mines in the vicinity of the measuring site. The paper provides evidence of the phenomenon and illustrates possible interpretations for its occurrence.

## 1 Introduction

In the frame of the HD(CP)$^2$ Observational Prototype Experiment (HOPE), the Raman lidar system BASIL was deployed and operated over a two-month period (April–May 2013) in the Atmospheric Supersite JOYCE, located within the Jülich Research Centre. This site is approximately 3 km West of the Hambach open-pit lignite mine, which represents the largest operational lignite mine on Earth, with a maximum depth of $\sim 200$ m (Fig. 1). The dump of this mine forms a large artificial hill, called Sophienhöhe, which reaches 302 m and is partially re-cultivated with forest. A second open-pit lignite mine, named the Inden mine, is located approximately 3 km south-west of the Supersite JOYCE.

The Hambach and Inden mines lie in the sectors 29–114° (red shaded area in Fig. 1) and 180–240° (blue shaded area in Fig. 1), respectively, relative to the location of the Raman

lidar. When wind blows from these directions, lignite particles from the two open-pit mines or the surrounding hill are lifted up from the ground and transported over the lidar site, with appreciable effects on the measurements.

Evidence of this particle transportation was found in lidar elastic backscatter echoes in a variety of case studies during HOPE, with the appearance of a specific odd feature in the upper portion of the convective boundary layer (CBL). Specifically, a minimum in lidar backscatter echoes at 532 and 1064 nm, with a backscatter reduction of approximately 10 % is observed. This feature is found to have a vertical extent of approximately 100 m and persist over a period of several hours, with an alternation of intensifications and attenuations of the phenomenon. Similar features with a comparable temporal duration and backscatter reduction had been reported by Sassen and Chen (1995) in the presence of light precipitation events; this phenomenon, referred to as *lidar dark band*, was demonstrated to be ascribable to changes in scattering properties of precipitating particles taking place during the snowflake-to-raindrop transition in the proximity of the melting level (Sassen et al., 2005; Demoz et al., 2000; Di Girolamo et al., 2003, 2012b).

Instead, the phenomenon reported in the present research effort appears in clear-air conditions and in the presence of strong convective activity within the boundary layer: we will refer to it in the following as the clear-air dark band phenomenon or the convective dark band phenomenon. In the following paper, we provide experimental evidence of this phenomenon and a possible physical interpretation for its occurrence.

The outline of the paper is as follows: Sect. 2 provides a description of the experimental set-up and a brief overview

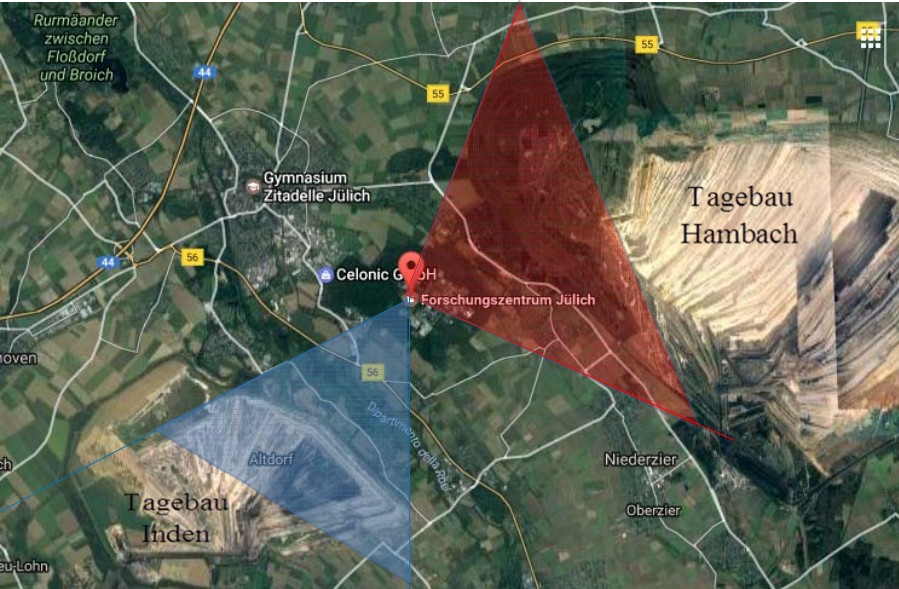

**Figure 1.** Locations of the Raman lidar system BASIL within the Jülich Research Centre (pink dot). The figure also indicates the location of the Hambach open-pit lignite mine (approximately 3 km East of the Research Centre), the large artificial hill Sophienhöhe (both within the angle cone 29–114°, red shaded area) and a second open-pit lignite mine (approximately 3 km south-west of the research centre, within the angle cone 180–240°, blue shaded area).

of the HOPE field campaign. Sect. 3 illustrates the measurements collected for a selected case study, providing remarks on the meteorological conditions occurring during these periods. Section 4 illustrates the hygroscopic and scattering properties of the sounded particles, while Sect. 5 formulates possible hypotheses for the interpretation of the observed phenomena. Finally, Sect. 6 summarises all results and provides some indications for possible future measurements and analysis.

## 2   BASIL and the HOPE field campaign

The University of Basilicata Raman lidar system (BASIL) is a ground-based Raman lidar hosted in a transportable "seatainer". BASIL performs high-resolution and accurate measurements of the vertical profiles of atmospheric temperature and water vapour, both in the daytime and at nighttime, exploiting both the rotational and vibrational Raman lidar techniques in the UV (Di Girolamo et al., 2004, 2006, 2009a, 2016; Bhawar et al., 2011). Aside from temperature and water vapour, BASIL also measures the vertical profiles of particle backscatter at 354.7, 532 and 1064 nm, as well as particle extinction and depolarisation at 354.7 and 532 nm (Griaznov et al., 2007; Di Girolamo et al., 2009b, 2012a, b). BASIL makes use of a neodymium-doped yttrium aluminum garnet (Nd:YAG) laser source, equipped with second and third harmonic generation crystals, which emits pulses at 354.7, 532 and 1064 nm. The receiver is built around a large aperture Newtonian telescope (primary mirror diameter: 0.45 m, focal length: 2.1 m) and two small-aperture telescopes (50 mm diameter lenses). The large aperture receiver incorporates eight channels for the detection of eight different signals (primarily Raman lidar signals), while the two small-aperture receivers include another three measurement channels for the detection of additional lidar signals. These eleven detected signals allow for the determination of the atmospheric variables listed above, plus additional ancillary parameters as the atmospheric boundary layer depth and the geometric (cloud base and top height, the latter in case of optically thin clouds) and optical (cloud optical depth for optically thin clouds) properties of clouds. More details on the experimental set-up of the system are provided in Di Girolamo et al. (2009a, 2017).

In this paper we illustrate measurements carried out in the frame of the High-Definition Clouds and Precipitation for advancing Climate Prediction (HD(CP)$^2$) Observational Prototype Experiment (HOPE, Macke et al., 2017). For the purposes of HOPE, BASIL was deployed in the Supersite JOYCE, located within the Jülich Research Centre (Central Germany, Lat.: 50°54′ N; Long.: 6°24′ E, Elev. 105 m). The system operated between 25 March and 31 May 2013, collecting more than 430 h of measurements distributed over 44 days and 18 Intensive Operation Periods (IOPs).

## 3   Results

The weather at the lidar site in Jülich on 18 April 2013 was characterised by the presence of clear sky conditions in

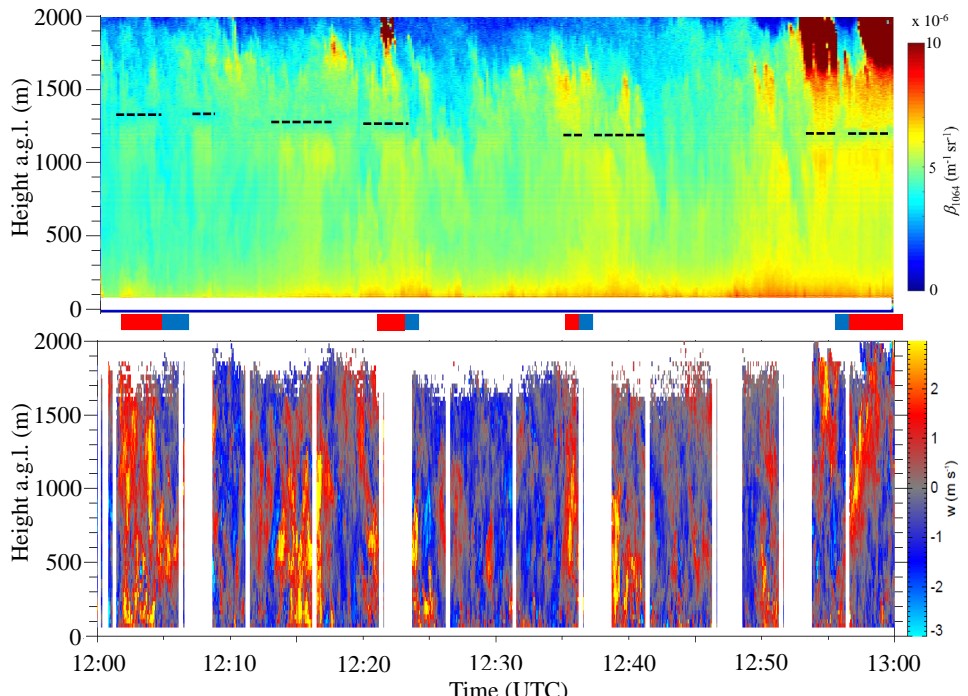

**Figure 2.** Time–height cross-section of $\beta_{1064}$ **(a)** and the vertical wind speed **(b)** in the time interval 12:00–13:00 UTC on 18 April 2013. The black dashed line in panel **(a)** around 1200 m highlights the presence of a persistent lidar backscatter reduction (clear-air dark band), with alternating intensity fluctuations. The red and blue areas in between the two plots indicate the up-draught and down-draught time intervals, respectively, identified in Fig. 6.

the morning until 06:00 UTC and by the passage of a cold front shortly afterwards. The passage of the cold front was followed by a circulation change from a south-westerly to west/north-westerly marine flow, with the sky clearing up in the late morning and the onset of a strong convective activity. Boundary layer clouds were found to form in the late morning and early afternoon, while broken cirrus clouds were observed throughout the day.

Figure 2 illustrates the time–height cross-section of the particle backscattering coefficient at 1064 nm, $\beta_{1064}$, as measured by BASIL (Fig. 2a), as well as the vertical wind speed, as measured by the University of Cologne wind lidar (Fig. 2b), in the time interval 12:00–13:00 UTC on 18 April 2013. For the purpose of these measurements, the two lidars were located within a distance of $\sim 80$ m. Figure 2a clearly reveals the presence of a significant aerosol loading within the boundary layer, which is tracing the presence of a well-mixed and quasi-stationary CBL at this time of the day, extending up to approximately 2000 m. The figure also reveals the presence of alternating up-draughts and down-draughts. The largest variability of $\beta_{1064}$ is observed in the interfacial layer, as a result of the penetration of aerosol-rich air rising from the ground and the entrainment of aerosol-poor air sinking from the free troposphere.

A persistent minimum in lidar backscatter is observed around 1200 m (black dashed line) with alternating inten-

sity fluctuations. This minimum persists albeit the clear presence of up-draughts (orange eddies, with positive vertical wind speed values) and down-draughts (blue eddies, with negative vertical wind speed values), and thus cannot be related to an aerosol layered structure in the mixing layer. Note that the backscatter minimum occurs preferably during up-draughts but not during down-draughts. This behaviour is clearly highlighted in Fig. 2, where the black dashed lines indicate that the lidar backscatter minima only appear in temporal coincidence with the vigorous up-draughts, which are testified by the positive vertical speed values (up to 2–3 m s$^{-1}$) measured by the wind lidar, but do not appear in coincidence with the down-draughts (negative vertical speed values down to $-2$–3 m s$^{-1}$). The presence of a persistent minimum in lidar backscatter at 1200 m, preferably during up-draughts, is also clearly visible in the particle backscattering coefficient data at 532 nm (not shown here). While we are concentrating on the time interval 12:00–13:00 UTC on 18 April 2013, additional evidence of this phenomenon was observed earlier and later in the day (i.e. 13:25–13:40, 13:50–14:05, 14:15–14:25, 14:35–15:00 UTC). The clear-air dark band phenomenon was also visible on other days (i.e. 20 April 2013) during HOPE, when the wind was blowing from directions overpassing the Hambach and Inden mines.

Figure 3 illustrates the vertical profile of $\beta_{1064}$ for the time interval 12:56:41–13:00:45 UTC on 18 April 2013 (4 min

average, green line), together with the vertical profiles of temperature, relative humidity (RH) and wind direction and speed, as measured by the radiosonde launched at 13:00 UTC from the nearby station of Hambach (4 km E-SE). The clear-air dark band is found to extend from 1150 to 1275 m with a vertical extension of 125 m and a minimum in particle backscattering at 1225 m (backscatter reduction is 8 %, corresponding to 0.4 dB). This band takes place few hundred metres below both the lifting condensation level (LCL, at 1725 m or 814 mbar) and the freezing level (at 1630 m or 823.2 mbar). The figure reveals that wind is blowing from directions in the interval from 265° (at surface) to 232°. More specifically, the particle backscattering reduction is located in the same height region (1125–1450 m) where wind is found to blow from directions in the interval 232–240°, i.e. from the directions where the Tagebau Inden open-pit lignite mine is located. In general, CBL wind direction measurements by radiosondes may be difficult to interpret as they may reflect rotations taking place within the convective plumes. However, wind direction values (236–242°) similar to those measured by the radiosonde are also present in the same height interval in the 1 h (12:00–13:00 UTC) average wind direction profile measured by the wind lidar (Fig. 4) with values throughout the whole profile from 270° (at surface) to 230° (at 1600 m).

Figure 3 also reveals that the air at this height is characterised by RH values in the range 60–62 %. Lignite particles advected by the wind to the lidar site are captured and ingested within the up-draughts and down-draughts associated with the intensive convective activity present at the lidar site. As a result of the adiabatic cooling associated with the uplift, air parcels undergo a sudden RH increase from values in the range of 60–62 % (environmental RH values at the base of the dark band) to values in the range of 75–80 % (these being the values reached within the lifting air parcel assuming an ideal adiabatic cooling with no air entrainment into the convective plumes or external air ingestion within the lifting air parcel). This sudden increase of RH has important effects on the size growth of the uplifted lignite aerosols.

Figure 5 illustrates the time–height cross-section of the particle depolarisation ratio at 532 nm, $\delta_{532}$, as measured by BASIL in the time interval 12:00–13:00 UTC on 18 April 2013, i.e. the same time interval considered in Fig. 2. Particle depolarisation ratio, defined as the power ratio of the cross-polarised to the co-polarised components of the particle backscattering coefficient, provides an indication of the degree of asphericity of sounded particles. Particle depolarisation depends not only on particles' shape, but also on their size and refractive index (among other, Burton et al., 2015). Water-coated aerosols, wet haze, fog, cloud droplets and small raindrops can be assumed to be almost spherical and are characterised by very small values of $\delta_{532}$, typically not exceeding 0.03. Low depolarising particles, usually smoke or urban aerosol, have depolarisation ratios between 0.03 and 0.1 (e.g. Burton et al., 2012), while high depolarising particles, such as desert or volcanic dust, have depolari-

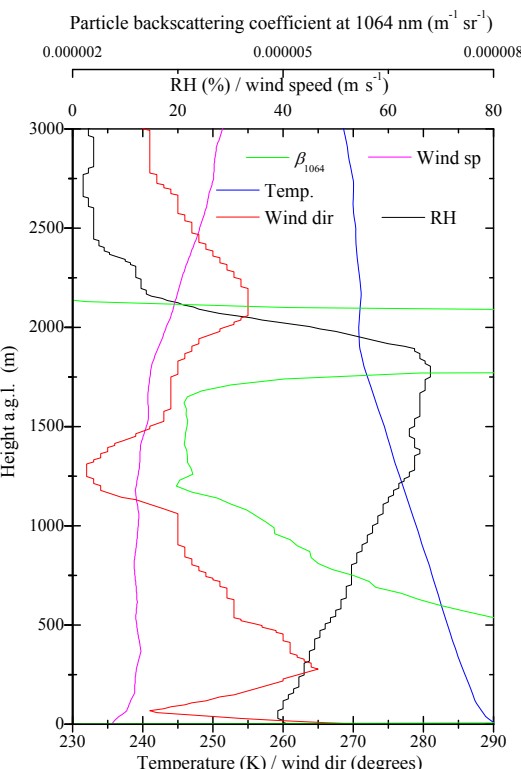

**Figure 3.** Vertical profile of $\beta_{1064}$ at 13:00 UTC on 18 April 2013 (12:56:41–13:00:45 UTC, green line), together with the vertical profiles of temperature (blue line), RH (black line), wind direction (red line) and speed (purple line) as measured by the radiosonde launched at 13:00 UTC from the nearby station of Hambach (4 km E-SE).

sation ratios that vary between 0.25 and 0.35 (e.g. Mona et al., 2012).

A proper calibration of particle depolarisation measurements requires accurate measurements of the cross-polarised and co-polarised components of the particle backscattering coefficient. However, accurate measurements of these quantities may be difficult to obtain, often as a result of the depolarising properties of different optical devices included in the receiver (Freudenthaler, 2016). This translates into a non-negligible uncertainty affecting particle depolarisation measurements, which includes both a systematic component (bias) and a random component (statistical error). For the present lidar system, these two components were estimated to be 10 % and 20 %, respectively (Di Girolamo et al., 2012a).

Figure 5 reveals a decrease in particle depolarisation at the same height and time intervals of the dark band. More specifically, $\delta_{532}$ decreases from values of 0.05–0.07 below the dark band to values of 0.02–0.03 within and above the dark band. A decrease of $\delta_{532}$ within and above the dark band is compatible with the conjectured size growth of the uplifted dry lignite particles, initially having a more irregular shape,

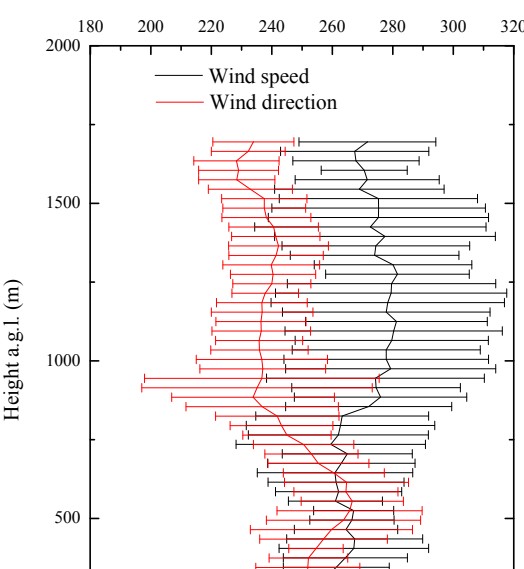

**Figure 4.** Vertical profile of wind speed and direction averaged over the time interval 12:00–13:00 UTC on 18 April 2013, as measured by the wind lidar located in the proximity of BASIL at the Supersite JOYCE. Profiles are reported with error bars, corresponding to ±1 standard deviation.

and then getting a more regular spherical shape as a result of the water uptake. Additionally, as previously observed for $\beta_{1064}$, the decrease of $\delta_{532}$ occurs during up-draughts, but not during down-draughts, as these latter values of $\delta_{532}$ are in the range of 0.02–0.04 both below and within the dark band. However, both below and within the dark band, values of $\delta_{532}$ are rather low, which is typical of aerosols including a large portion of carbonaceous species as in those resulting from fossil fuel combustion that have a rather spherical shape (Dieudonné et al., 2017; Müller et al., 2007). Particle depolarisation ratio measurements, while providing some information on particle shape, may also be used for aerosol typing and mass concentration studies (among others, Petzold, 2011; Burton et al., 2012).

The presence of the clear-air dark band phenomenon during up-draughts is also well documented in Fig. 6, as it illustrates the simultaneous vertical profiles of $\beta_{1064}$, $\delta_{532}$ and RH as measured by BASIL, and the wind direction and vertical wind speed as measured by the wind lidar for a number of consecutive up-draught and down-draught time intervals. Sharp lidar backscatter minima are only observed around 1200 m in temporal coincidence with positive vertical speed

values (Fig. 6a, c, e, g), while slowly variable backscatter values are observed at these heights in temporal coincidence with negative vertical speed values (Fig. 6b, d, f, h). Wind direction values in the time intervals and vertical regions characterised by the presence of backscatter minima are very similar to those observed in these vertical regions during the down-draught periods. This observation supports the hypothesis that the observed backscatter minima are not caused by the presence and sounding of different types of particles that might originate from different aerosol sources, as sounded air masses are coming from the same direction both during up-draughts and down-draughts. However, wind direction measurements by Doppler wind lidar require a minimum integration time of 5 min, as in fact a number of off-zenith measurements are needed to determine the horizontal wind component. This implies that a perfect time matching between BASIL measurements of $\beta_{1064}$ and RH and wind lidar measurements of wind direction was not possible in Fig. 6, as in fact the integration time for BASIL measurements was taken as coincident (within 10 s, which is the maximum time resolution for BASIL measurements) with the duration of the up-draughts and down-draughts, typically lasting 1–2 min, while the 5 min integration time wind direction measurements may superimpose to consecutive up-draughts and down-draughts. Additionally, the approach of determining wind directions by Doppler wind lidar measurements is affected by a large uncertainty (typically around 25° in the vertical regions characterised by the presence of backscatter minima).

Similar considerations apply for RH measurements. Accounting for the error affecting these measurements, RH values, observed by BASIL in the altitude region where backscatter minima take place, have very similar values sounded during the up-draught and down-draught, which would support the hypothesis of the presence of a reversal (evaporation) process in the down-draughts, which instead is not observed. Vertical profiles of RH are obtained from water vapour mixing ratio and temperature profile measurements by BASIL, which are based on the application of the vibrational and rotational Raman lidar technique, respectively. Both techniques rely on Raman backscatter phenomena characterised by cross-sections that are several orders of magnitude smaller than the elastic backscatter cross-section. This makes the water vapour mixing ratio and temperature measurements, and consequently RH measurements, very difficult to perform, especially in daytime around noon, as is the case for the measurements illustrated in this paper, as a result of the large solar irradiance affecting the measurements during this portion of the day. This translates into a large statistical uncertainty affecting RH measurements with a random error of 4–8 % (error bars in Fig. 6) in the altitude region ($\sim$ 1200 m) where the particle backscatter minima are observed.

Clear-air dark bands were mostly observed in the absence of a cloud topped CBL. However, few clouds were observed for this specific case study in the upper portion of the CBL

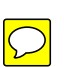
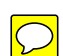

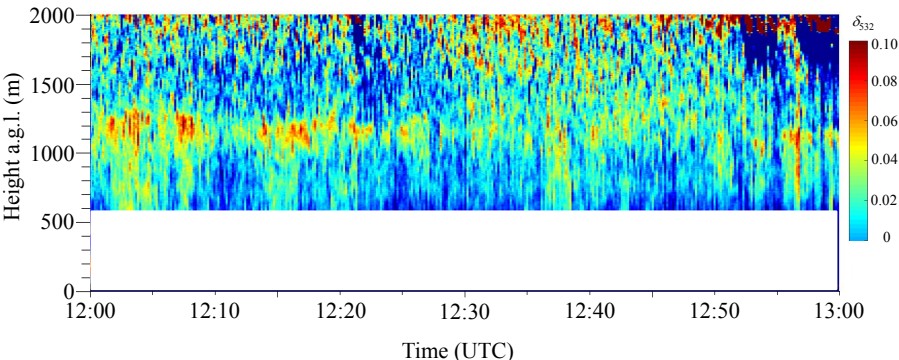

**Figure 5.** Time–height cross-section of particle depolarisation at 532 nm in the time interval 12:00–13:00 UTC on 18 April 2013.

**Figure 6.** Vertical profiles of $\beta_{1064}$, $\delta_{532}$ and RH (as measured by BASIL) and the wind direction and vertical wind speed (as measured by the wind lidar) for eight consecutive up-draughts and down-draughts time intervals during the time period 12:00–13:00 UTC on 18 April 2013: **(a)** 12:01:34–12:04:37 UTC, **(b)** 12:04:37–12:06:39 UTC, **(c)** 12:21:01–12:23:03 UTC, **(d)** 12:23:03–12:24:04 UTC, **(e)** 12:35:17–12:36:18 UTC, **(f)** 12:36:18–12:37:19 UTC, **(g)** 12:56:41–13:00:45 UTC and **(h)** 12:55:39–12:56:41 UTC). Green-dashed ellipses highlight dark band features during the up-draught intervals. The up-draught and down-draught time intervals considered in the present figure are identified in Fig. 2 with red and blue areas.

at 12:53–13:00 UTC (orange-brown features in Fig. 2a and strong backscattering enhancement observed above 1600 m in Fig. 3). The occurrence of these clouds is discussed in more detail in the final portion of Sect. 4.

## 4 Hygroscopic and scattering behaviour of lignite particles

Aerosol particles can be classified according to their affinity for water as hygroscopic, neutral or hydrophobic. The characterisation of particle hygroscopicity is of primary importance in climate monitoring and prediction. Model studies have demonstrated that RH has a critical influence on aerosol climate forcing (Pilinis et al., 1995), with hygroscopic growth at large RH values having important implications in terms of aerosol direct effect (Wulfmeyer and Feingold, 2000).

Lignite, often referred to as brown coal, is a combustible sedimentary rock formed from naturally compressed peat with a carbon content around 60–70 %. The high moisture content of lignite (approximately 50–60 %) is an undesirable inert component, which significantly reduces its calorific value. Consequently, when employed in conventional power plants, a considerable portion of lignite's energy content is typically required prior to combustion to evaporate this large portion of water. For this reason, following the mining process, raw lignite usually undergoes effective drying processes. This is indeed the case for the two open-pit lignite mines of Hambach and Inden in the proximity of the lidar station, where a drying process based on the pulverisation of the lignite particles is applied.

Dried lignite particles produced in open pit lignite mines have a very marked hygroscopic behaviour (Schobert, 1995; Krawczykowska and Marciniak-Kowalska, 2012) and, as a result of this behaviour, effectively absorb moisture from the atmosphere. Measurements of the particle size distribution of lignite particles escaped from heavy industrialised areas (mining and power stations operations) in the form of fly ash or fugitive dust have been reported by several authors (among others, Triantafyllou et al., 2006; Civiš and Hovorka, 2010). Specifically, Triantafyllou et al. (2006) were able to measure the particle size distribution of fly ash injected into the atmosphere from elevated stacks in power stations, thus identifying a prominent particle mode at $\sim 8 \, \mu m$, with approximately 80 % of the particles smaller than 10 µm. Civiš and Hovorka (2010) reported size distribution measurements for brown coal with an average particle size of 1.84 µm. All of these authors revealed a limited degree of poly-dispersion of atmospheric lignite particles. When considering a log-normal size distribution, the degree of poly-dispersion or width of the particle size distribution is expressed in terms of the percentage standard deviation of the logarithm of the distribution, $\sigma$. Narrow size distributions for brown coal particles, with values of $\sigma$ in the interval 5–10 %, have been reported

by a variety of authors (Mujuru et al., 2009; Civiš and Hovorka, 2010; Wang and Tichenor, 1981).

The solution effect typically dominates hygroscopic particles' growth when the radius is small (smaller than the critical radius $r_c$), which results in small solution droplets being in equilibrium with water vapour at RH values less than 100 % (Yau and Rogers, 1989). At this stage, small increases in RH determine particles' size growth until equilibrium is newly reached. This mechanism is possibly responsible for the lignite particle growth below the LCL, ultimately leading to the appearance of a minimum in lidar backscatter echoes (i.e. the above mentioned clear-air dark band phenomenon). The increase in particles' radius associated with the relative humidity change experienced by the adiabatically uplifted air parcel can be estimated based on the application of the Köhler equation. When RH values are smaller than 100 %, the Köhler equation is dominated by the solution term, which depends on the mass and molecular weight of the solute species and the so called van't Hoff factor. Based on literature values of these quantities, the above specified increase of RH from 60–62 to 75–80 % would result in a particle size growth in radius by 10–20 %. In this study, we are considering an initial size for the dry lignite particles of 1.84 and 8 µm, as reported by Civiš and Hovorka (2010) and Triantafyllou et al. (2006), respectively.

Scattering properties of lignite particles have been simulated based on the application of a light scattering code for spheres based on Mie theory (http://philiplaven.com/mieplot.htm, last access: 28 December 2017). In this respect, the small values of $\delta_{532}$ characterising the observed aerosol particles call for a very limited degree of asphericity, which makes Mie theory still successfully applicable for the simulation of particles' scattering properties (Martin, 1993; Mishchenko and Lacis, 2003). In order to properly simulate the scattering processes, accurate information on particle refractive index are required, beyond those on particle size distribution already provided above. Accurate measurements of lignite refractive index were reported by Lohi et al. (1992), who observed values of the real and imaginary part of the complex refractive index of .70 and $1 \times 10^{-6}$, respectively. Similar values were reported by McCartney and Ergun (1962) and Read (2008). Simulations of the scattering properties of lignite particles are illustrated in Fig. 7. The figure shows the variability of the quantity $Q_{back} \times r^2$ as a function of $r$, with $Q_{back}$ being the backscattering efficiency and $r$ the particle radius. These simulations are obtained by considering a log-normal size distribution with a value of $\sigma$ of 5 %. The simulation in Fig. 7a considers a minimum radius of 1.84 µm, as measured by Civiš and Hovorka (2010), while the simulation in Fig. 7b considers a minimum radius of 8 µm, as measured by Triantafyllou et al. (2006). Both simulations consider a sounding wavelength of 1.064 µm, which is the laser wavelength used for the dark band lidar measurements illustrated in Fig. 2. The quantity $Q_{back} \times r^2$ represents the single-particle backscattering coefficient, assum-

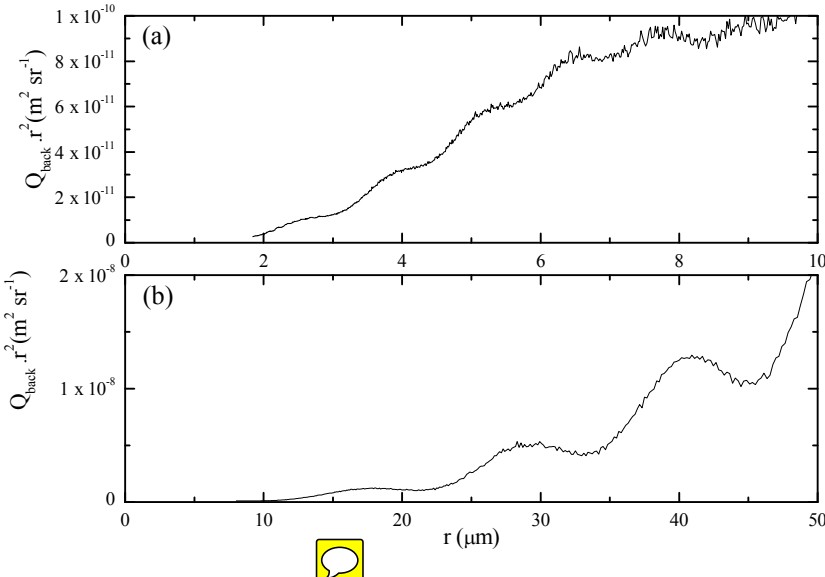

**Figure 7.** Simulations of the ~~backscattering efficiency~~ as a function of particle radius for lignite particles. Simulations consider a log-normal size distribution with a percentage standard deviation of 5 %. **(a)** Selection of a minimum radius of 1.84 μm, as given by Civiš and Hovorka (2010) and a maximum radius of 10 μm; **(b)** selection of a minimum radius of 8 μm, as given by Triantafyllou et al. (2006) and a maximum radius of 100 μm. Both simulations consider a sounding wavelength of 1.064 μm.

ing a constant particle number density $n$. Figure 7 reveals the presence of marked oscillations in particle backscattering efficiency. As a result of these oscillations, for specific radius values of the dry lignite particles (for example, 6.5, 7.5, 18, 28.5 or 41 μm), a reduction in $Q_{back} \times r^2$ of 8–27 % (0.35–1.4 dB) is observed for a particle size growth by 10–16 %, which is compatible with the size growth experienced by these particles during their adiabatic ascent. Thus, we believe that the observed dark band phenomenon is associated with the oscillations in the particle backscattering coefficient, ultimately leading to Mie back-scattered signal intensity fluctuations. These backscattering coefficient oscillations are to be attributed to the limited degree of poly-dispersion of atmospheric lignite particles. It is to be specified that these oscillations smooth down and finally disappear in the case of larger values of $\sigma$, thus wider particle size distributions (distributions with a higher degree of poly-dispersion) are considered. This is clearly highlighted in Fig. 8, which illustrates the simulated values of $Q_{back}$ for lignite particles as a function of particle radius, considering a log-normal size distribution with different values of $\sigma$ (0.1, 5, 10 and 20 %) again considering a sounding wavelength of 1.064 μm. As for Fig. 7, the simulation in panel (a) considers a minimum radius of 1.84 μm, as measured by Civiš and Hovorka (2010), while the simulation in panel (b) considers a minimum radius of 8 μm, as measured by Triantafyllou et al. (2006). The figure clearly reveals that both in the smaller and larger particle domains, the consideration of progressively larger values of $\sigma$ leads to a progressive smearing down of the $Q_{back}$ oscilla-

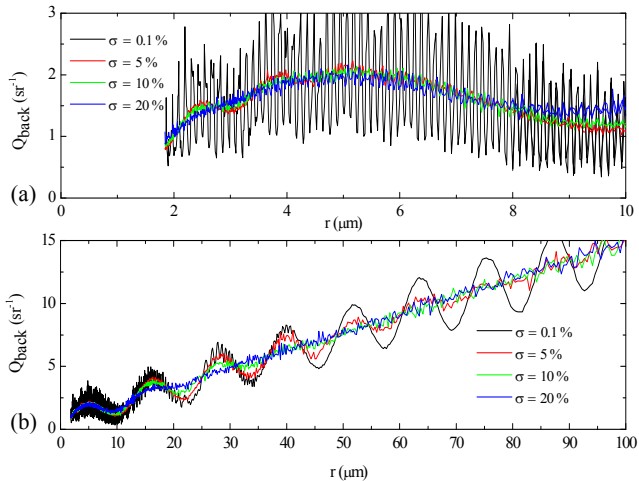

**Figure 8.** Simulations of the backscattering efficiency at 1064 nm as a function of particle radius for lignite particles, considering a log-normal size distribution with values of $\sigma$ equal to 0.1, 5, 10 and 20 %. **(a)** Selection of a minimum radius of 1.84 μm, as given by Civiš and Hovorka (2010) and a maximum radius of 10 μm; **(b)** selection of a minimum radius of 8 μm, as given by Triantafyllou et al. (2006) and a maximum radius of 100 μm. Both simulations consider a sounding wavelength of 1.064 μm.

tions, which are still present for values of $\sigma \leq 10$ % but are almost absent for $\sigma = 20$ %.

An additional quantity, namely the backscatter colour ratio, BCR, i.e. the ratio of total backscattering coefficients at 1064 and 532 nm, was determined from BASIL measure-

ments. Colour ratio profiles measured during the time interval considered in the present study (12:00–13:00 UTC on 18 April 2013, not shown here) indicate values in the range 0.40–0.45 below the dark band and in the range 0.33–0.36 within the dark band region. The colour ratio decrease is an indication of the increase of particle size. This represents an additional experimental evidence of the conjectured particles' growth, which represents the basis of the given interpretation of the observed phenomenon. Furthermore, small backscatter colour ratio values, as ~~with~~ those found both below and within the dark band, are indicating relatively large particles (Burton et al., 2013), compatible with those conjectured in the present study and presently considered in our simulations. The variability of backscatter colour ratio as a function of particle radius has been simulated with the same Mie scattering code already used above, with simulations revealing that values of BCR in the range of 0.33–0.45 are compatible with particle size in the range of 7–11 μm. Finally, backscatter colour ratio values in the range of 0.33–0.45 combined with values of $\delta_{532}$ in the range of 0.02–0.07 are in agreement with previously observed values of these quantities as reported by a variety of authors (de Villiers et al., 2010: BCR = 0.3–0.5 and $\delta_{532}$ = 0.02–0.08; Burton et al., 2014: BCR = 0.55 and $\delta_{532}$ = 0.07; Burton et al., 2015: BCR = 0.47 and $\delta_{532}$ = 0.06–0.09). Similar values (BCR = 0.35–0.54 and $\delta_{532} < 0.05$) were also reported by Franke et al. (2003) and Müller et al. (2007) for Southeast Asian aerosols, which were argued to possess a pronounced coarse mode with large particles originating mainly from coal and dried plants used for domestic heating and cooking (Müller et al., 2007).

The comparison of simulated values of single-particle backscattering coefficient $Q_{back} \times r^2$ ($\sim 3 \times 10^{-11}$ m$^2$ sr$^{-1}$ for a particle radius of 4 μm and $\sim 1 \times 10^{-9}$ m$^2$ sr$^{-1}$ for a particle radius of 20 μm) with measured values of the volume backscattering coefficient $\beta_{1064}$ (in Fig. 6, in the range 2.5–3.5 $\times 10^{-6}$ m$^{-1}$ sr$^{-1}$ within the dark band) leads to an estimate of particle number density $n$ of 0.8–1.2 $\times 10^5$ and 2.5–3.5 $\times 10^3$ m$^{-3}$ in the small and large particles' domain, respectively. These values of $n$ are in agreement with literature values for continental and urban polluted aerosols (e.g. Mészáros, 1991; 0.8–3.5 $\times 10^5$ m$^{-3}$ for a particle radius of 4 μm and 1–2 $\times 10^3$ m$^{-3}$ for a particle radius of 20 μm).

The solution effect growth of particles to equilibrium size associated with increasing RH can be continued up to a RH value of 100 % and slightly beyond. Cloud formation at the top of the CBL will finally take place above the LCL if the critical saturation ratio, $S_c$, corresponding to the peak of the Koehler curve, is reached. $S_c$ is typically reached for supersaturation values of 0.5–1 %, depending on the composition (and consequently the level of hygroscopicity) and size of the aerosol particle acting as condensation nuclei. In the case of lignite particles, typical values of $S_c$ and of the critical radius, $r_c$, are in the range 0.5–1 % and 1–10 μm, respectively. Up to this point RH had to be increased in order for the droplet

to grow. However, if RH slightly exceeds $S_c$, the particle is enabled to grow beyond $r_c$ and its saturation ratio falls below $S_c$. As a consequence, the water vapour condensates on the droplet, which will continue to grow without the need for a further increase in saturation ratio (Yau and Rogers, 1989). When this occurs, clouds can form on the top of the CBL. These processes are responsible for the clouds observed in the upper portion of the CBL at 12:53–13:00 UTC (orange-brown features in Fig. 2a). In the clouds, the droplet growth process does not continue indefinitely as many droplets are present and all of them compete for the same available water vapour.

## 5 Discussion of the observed phenomena

Raman lidar measurements illustrated in this paper reveal the presence of a persistent minimum (dark band) in lidar elastic backscatter echoes in the upper portion of the CBL. This phenomenon appears in clear sky conditions, in the presence of strong convective activity, and is mostly confined to up-draughts. Adiabatic cooling within the up-draughts leads to an RH increase and consequent particle growth, especially in hygroscopic particles. If we assume that most of the particles we observe are dry hygroscopic lignite particles from the surrounding lignite open-pit mines and that their size distribution is mono-disperse or very narrow, we must conclude that the observed dark band is related to the oscillations of the backscatter efficiency as described by Mie-theory, ultimately leading to intensity fluctuations of the Mie back-scattered radiation. In the presence of a wider particle size distribution the backscatter oscillations should smear out, if not disappear. This interpretation is also supported by the outcome of the lidar depolarisation measurements. In fact, water uptake by uplifted dry lignite aerosols, initially having a more irregular shape, confers a more regular spherical shape to these particles, this shape change being responsible for the decrease in particle depolarisation observed at the same height and time intervals of the dark band (Fig. 6), again mostly confined to up-draughts.

The fact that the dark band and the depolarisation decrease are confined to the up-draughts can be explained in two ways: either the adiabatic warming, and the consequent decrease in RH, in down-draughts does not lead to an inversion of the particle growth (i.e. there is a hysteresis, and humidified particles do not evaporate the water amount they incorporated during their way up) or down-draughts transport different and/or modified particles to the up-draughts. These particles might be less hygroscopic and thus change their size less with RH. The possibility that particles within the down-draughts are different from those within the up-draughts increases in the interfacial layer due to ~~the~~ entrainment effects and is possibly testified by the presence of smaller particle backscatter values within the down-draughts with respect to those observed within the up-draughts (see Fig. 2). This is

possibly associated with the entrainment of air from the free-troposphere at the top of the CBL, which may ultimately lead to changes in particle size distribution and scattering properties. Evidence of the sharp entrainment of air pockets from the free troposphere into the boundary layer, which gradually mix with the environmental air, has been reported by a variety of authors (Couvreux et al., 2005, 2007, Wulfmeyer et al., 2010, 2016; Turner et al., 2014). Particle size distribution within the down-draught could be not as narrow as in the up-draughts, resulting in a smear out of backscatter efficiency oscillations.

The hygroscopic growth of particles is dependent upon aerosol composition and may be subject to monotonic (smoothly varying) or deliquescent (step change) growth. A dry hygroscopic aerosol transforms into a solution droplet when RH increases beyond the so called deliquescence point. Particle deliquescent growth, as with the one characterising lignite particles (Brooks et al., 2004), shows a hysteresis behaviour during the uptake and loss of water, i.e. exhibits difference values for the deliquescence and efflorescence relative humidity (Sjogren et al., 2007); this hysteresis behaviour ultimately determines a less efficient evaporation process (Seinfeld and Pandis, 2006). More specifically, when RH decreases, the solution droplet starts reducing in size through the evaporation of the previously taken up water at the efflorescence point, which is found at a much lower RH value than the deliquescence point (Oatis et al., 1998).

## 6   Summary and final remarks

This paper illustrates measurements carried out by the Raman lidar system BASIL in the frame of HOPE, revealing the presence of a persistent minimum in clear-air backscatter echoes in the upper portion of the convective boundary layer. Backscatter reduction is approximately 10 %, has a vertical extent of approximately 100 m and persists over a period of several hours. We refer to this phenomenon as to the clear-air dark band or the convective dark band. This has to be distinguished from a similar phenomenon, with comparable temporal duration and backscatter reduction, observed in the presence of light precipitation events (Sassen and Chen, 1995; Demoz et al., 2000), the so called lidar dark band, ascribable to the changes in precipitating particles' scattering properties taking place during the snowflake-to-raindrop transition.

Dark bands illustrated in this paper are observed in the presence of strong convective activity within the boundary layer, when dry lignite aerosol particles are advected from the surrounding open pit mines, the bands are mostly confined to the convective up-draughts. The phenomenon is interpreted as being related to the oscillations characterising lignite particle backscatter efficiency, ultimately leading to Mie back-scattered signal intensity fluctuations. These backscatter efficiency oscillations are attributed to the limited degree of poly-dispersion and the high hygroscopicity of atmospheric lignite particles. Adiabatic cooling within the up-draughts leads to an RH increase and consequent particle growth.

Adiabatically warming and thus a decrease in RH in down-draughts does not lead to an inversion of the particle growth and humidified particles do not or only partially evaporate the water they took up during the up-draught. Additionally, down-draughts may transport different particles than the up-draughts. These are possible motivations for having clear-air dark bands are mostly confined to up-draughts. Observations and results illustrated in this paper support the interpretation of the phenomenon as a purely microphysical growth mechanism; however, the possibility that other mechanisms (e.g. dynamics) may also participate and contribute to the appearance of the phenomenon cannot be completely excluded.

*Data availability.* Data used in this study, together with the related metadata, are available from the public data repository HD(CP)$^2$ Data Archive (Stamnas et al., 2016), which is freely accessible by all users from the HD(CP)$^2$ Web Portal (http://icdc.zmaw.de/1/projekte/hdcp2.html TS2). The details for the data structure and organization can also be found in Stamnas et al. (2016).

*Competing interests.* The authors declare that they have no conflict of interest.

*Acknowledgements.* Measurements illustrated in this paper were supported on the basis of a specific cooperation agreement between Scuola di Ingegneria – Università degli Studi della Basilicata, Leibniz Institute for Tropospheric Research and the Max Planck Institute. We also wish to thank Dario Stelitano, from Scuola di Ingegneria – Università degli Studi della Basilicata, for his support during the HOPE field deployment.

Edited by: Matthias Tesche
Reviewed by: three anonymous referees

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

## Remarks from the typesetter