# Peer review of "Clear-air lidar dark band"

_Atmospheric Chemistry and Physics, 2017_

## Referee Comment (RC1) · Anonymous Referee #2 · 18 Dec 2017

**General Comments**

This paper is very well written and deals with a unique lidar aerosol observation. The case is made for subtle growth by condensation of water on a narrow size distribution of aerosol. Under very specific atmospheric conditions the growth can result in a Mie backscatter minimum at a certain altitude. The lidar and radiosonde data are brought together to make a consistent argument for being able to see this occurrence. This paper is appropriate for ACP and can be published with minor corrections.

**Specific comments**

Abstract: no comments

Page 2: Line 23: These eleven detected signals allow(s) determining . . .

Page 3: Line 15: This minimum persist(s) albeit . . .

[Figure]

Page 5: Line 1: Figure 7 reveals . . . Shouldn't that be Figure 6?

Figure 7: end of caption: 1.064 micrometers not millimeters

---

## Referee Comment (RC2) · Anonymous Referee #1 · 21 Dec 2017

Summary:

The paper reports on an aerosol event that was observed with lidars in southwest Germany during the HOPE campaign in 2013. Over the presented period of 1 hour, the multi-parameter BASIL lidar, the key instrument in this study, measured a slowly descending, geometrically thin and stable filament of boundary-layer aerosols that exhibited diminished elastic light backscattering. This feature, which the authors dub a clear-air lidar dark band, contrasts with the prevalent dynamic conditions. With the help of wind data from a near-by wind lidar and radiosonde data it is argued that the optical phenomenon was produced by lignite particles transported from an open-pit mine about 3 km away, and that it occurred in updrafts rather than downdrafts at a background relative humidity of about 62 percent. After a short literature survey on lignite particle emissions, the authors then employ Mie theory to model the backscattering efficiency of lignite particle spectra with relatively narrow size distributions and

conclude that the observed lidar dark band may be the result of the particles growing by water uptake during updrafts, with the backscattering efficiency passing through a local minimum. No satisfactory explanation is given why the reversal process likely to occur in downdrafts does not produce a similar effect.

The subject material falls within the scope of Atmos. Chem. Phys., and is of interest to the aerosol lidar and modeling communities. The presented experimental data are interesting, and the explanation is plausible, however, more effort should be made to better support the conclusions, especially, profiles of other parameters as measured wit BASIL should be included in the study, and the origin of the observed air masses should be accessed more carefully.

In summary, the manuscript is suited for publication in Atmos. Chem. Phys., however, revisions are deemed necessary.

General comments:

1. BASIL is, according to Section 2, a high-performance multi-parameter instrument, capable of measuring water vapor, temperature, and several aerosol optical properties at up to three wavelengths, including depolarization ratio and extinction ratios. How come then that only its range-corrected backscatter signal at 1064 nm (RCS1064) and the 532-nm depolarization ratio (DR532) are used to visualize the lidar dark band? Neither the water vapor and temperature measurements are utilized to determine relative humidity directly nor the set of particle optical properties is exploited to obtain some microphysical parameters of the aerosol as constraints for the model runs. And why is only RCS1064 given and not the particle backscatter coefficient (PBC1064), the physically meaningful quantity? The reviewer understands that the measurement conditions were certainly difficult around noon, so probably water vapor and temperature may not be readily available. But some sort of microphysical analysis and certainly calculation of PBC1064 should be possible and should be part of this investigation.

2. Because DR532 is significant, application of Mie theory is questionable. Please,

discuss.

3. Discussion on Figs. 7 and 8, page 6: Why is backscattering efficiency in arbitrary units, and why is backscattering efficiency presented at all? Assuming a constant particle number density n, particle backscatter coefficient is proportional to the product of backscattering efficiency Q and the square of the particle radius, $r^2$. So, $Qr^2$ should be modeled to explain the lidar dark band, and ideally it should be compared to the BASIL measurements of PBC1064. Note that a decrease in Q of 10% (top panel) and 40% (bottom panel) as modeled maximally for small and large initial lignite particles (Fig. 7) would be compensated for by an increase in r by a factor of 1.05 and 1.19, respectively.

4. According to Figs. 3 and 4, westerly winds (> 240 degrees wind direction) prevailed throughout the boundary layer which did not pass over the open-pit mine (Fig. 1). The only exception is the dark-band layer characterized by slightly more southerly winds blowing from the edge of the pit. So isn't it possible that BASIL simply observed air masses of different origin (and therefore different aerosols) at different heights? Or asked differently, how certain can one be that really aerosol growth was observed? Please, discuss.

5. Page 7, lines 29ff., page 8, lines 27ff.: In their discussion of the differing aerosol behavior in up- and downdrafts the authors mention the possibility that different particles were measured. The reviewer agrees, see 4. (above). To investigate this important issue, the authors should not only show vertical wind speed for up- and downdraft periods in Fig. 5 but also wind direction.

Specific comments:

1. Page 3, lines 29, 30: Please, discard RCS1064 in Fig. 3 and show the full profile of PBC1064 instead for a better understanding of the measurement situation.

2. Page 4, line 25: This is no true. Depolarization ratio also depends on size and

refractive index of the aerosol particles. Please, be more accurate.

3. Page 5, line 1: The authors probably mean 'Figure 6'. The observation made is not at all obvious. Please, show some example profile pairs (PBC1064, DR532).

4. Page 6, line 4: How do the authors arrive at the conclusion of 15-30% size growth? And is it volume, mass, diameter growth?

Figures:

All figures: Use the same style, make sure axis labels and titles are easy to read.

Figure 2, upper panel: Show PBC1064. Dotted line hard to see. Color bar?

Figure 2, lower panel: Up-/down drafts hardly visible, use different color table.

Figure 3: Show PBC1064, full profile.

Figure 4: Use colored curves instead of symbols, explain curves in caption. Choose narrower direction range so that relevant values can be better judged, for instance 180-320 degrees.

Figure 5: Good start. Please, show PBC1064 instead of RCS1064. Also include DR532 and wind direction for a better characterization of the particles and the measurement conditions, respectively.

Figure 6: As previously noted, it is hard to discern (anti)correlations comparing Figs. 2 and 6 by eye. Example profile pairs (PBC1064, DR532) would help.

Technical corrections:

1. Throughout text: Subscripts that are not variables must not be italic

2. Abstract, 1st line: Remove ')'at line end

3. Page 1, line 17: '200 m'

4. Page 2, line 8: 'for a selected'

5. Page 3, line 25: 'on other days'

6. Page 4, line 8: 'wind to the'

7. Reference Civis...: Remove 'Jan'

8. Reference Krawczy...: Include 'and' between authors.

9. Reference Yau...: Include 'and' between authors, move to end of list

---

## Referee Comment (RC3) · Anonymous Referee #3 · 22 Dec 2017

Summary:

This paper presents an interesting case study of multiwavelength lidar measurements (including linear depolarization ratio measurements) of an unusual aerosol event made during the HOPE campaign in 2013. A thin layer of boundary layer aerosol with an apparently distinctly low associated backscatter coefficient was observed giving rise to a "clear-air dark band". The authors discuss the measurements and offer an explanation for the observations. Namely, they hypothesis that the layer was produced by hydrating lignite particles from a local open-pit mine.

In the main, the paper is clear, the measurements appear sound and the offered explanation seems plausible. The paper is suitable for ACP. However, there are a few areas that should be addressed before publication.

While preparing my review I have noticed that Anonymous Reviewer #1 has posted

their review, which seems thorough. In the interest of efficiency (and the coming holidays) I will frame my review with reference to Reviewer #1s' comments.

General comments:

I can state that I agree with the general comments of Reviewer#1 with the exception of the contention that "No satisfactory explanation is given why the reversal process likely to occur in downdrafts does not produce a similar effect". I find that the discussion offered in the Section 5 of the paper is plausible and "complete enough" in the context of the present work. Perhaps the authors can state that this is a preliminary hypothesis and outline what exact work would be needed to test their offered explanation in a quantitative manner.

I can also state that I especially agree with Reviewer#1 that the Figures need to be, in general, improved.

Specific Additional Comment

-Line 30 on Page 4 refers to Figure 6: I think this should be Figure 5 OR the authors have left a figure out by mistake. The discussion makes me think they are referring to a line-plot of the depolarization ratio vs height. If this is not the case, and they are really mean to be referencing Figure 5, then they should consider inserting a line plot of the depolarization. In any case, Figure 5 is not clear enough for me to draw any quantitative information from !

---

## Author Comment (AC1) · 28 Feb 2018

Dear Editor,

We are very grateful to the three referees for their appropriate and constructive suggestions and for their proposed corrections to improve the paper. We have addressed all issues raised and have modified the paper accordingly. If you and the referee agree on that, we are also ready to submit a revised version of the paper where all these changes have been introduced. We believe that, thanks to their precious inputs, the manuscript has now sensitively improved. Below is a summary of the changes we made and our specific responses to the referees' comments and recommendations.

**Summary of the changes**

**(in black is the original comments of the referee and in red our responses)**

**Anonymous Referee #1**

Summary:
The paper reports on an aerosol event that was observed with lidars in southwest Germany during the HOPE campaign in 2013. Over the presented period of 1 hour, the multi-parameter BASIL lidar, the key instrument in this study, measured a slowly descending, geometrically thin and stable filament of boundary-layer aerosols that exhibited diminished elastic light backscattering. This feature, which the authors dub a clear-air lidar dark band, contrasts with the prevalent dynamic conditions. With the help of wind data from a near-by wind lidar and radiosonde data it is argued that the optical phenomenon was produced by lignite particles transported from an open-pit mine about 3 km away, and that it occurred in updrafts rather than downdrafts at a background relative humidity of about 62 percent. After a short literature survey on lignite particle emissions, the authors then employ Mie theory to model the backscattering efficiency of lignite particle spectra with relatively narrow size distributions and conclude that the observed lidar dark band may be the result of the particles growing by water uptake during updrafts, with the backscattering efficiency passing through a local minimum. No satisfactory explanation is given why the reversal process likely to occur in downdrafts does not produce a similar effect.

With the support of the additional observations from BASIL and the wind lidar now introduced in the paper, an integration in the interpretation of the observed phenomena, especially of the absence of a reversal process in the downdrafts, is provided both in this response to the referee and in the revised version of the paper (see specific details on this aspect in the answers below).

The subject material falls within the scope of Atmos. Chem. Phys., and is of interest to the aerosol lidar and modeling communities. The presented experimental data are interesting, and the explanation is plausible, however, more effort should be made to better support the conclusions, especially, profiles of other parameters as measured with BASIL should be included in the study, and the origin of the observed air masses should be accessed more carefully.

We agree with the referee on the need to support the characterization and interpretation of the observed phenomenon based on the consideration in the study of the vertical profiles of other parameters measured by BASIL. In this direction, vertical profiles for a variety of additional parameters (namely, particle backscattering coefficient at 1064 nm, particle depolarization at 532 nm and relative humidity from BASIL, and wind direction from the wind lidar) in temporal coincidence with updrafts and downdrafts are now considered in the study and have been introduced in the modified version of figures included in the revised paper. The combined use of these additional information also allows to get a further confirmation of the origin of the observed air masses (see details below).

In summary, the manuscript is suited for publication in Atmos. Chem. Phys., however, revisions are deemed necessary.

We appreciate the possibility for improve the manuscript based on the precious comments and suggestions provided by the reviewer.

General comments:

1. BASIL is, according to Section 2, a high-performance multi-parameter instrument, capable of measuring water vapor, temperature, and several aerosol optical properties at up to three wavelengths, including depolarization ratio and extinction ratios. How come then that only its range-corrected backscatter signal at 1064 nm (RCS1064) and the 532-nm depolarization ratio (DR532) are used to visualize the lidar dark band? Neither the water vapor and temperature measurements are utilized to determine relative humidity directly …

The vertical profiles of water vapour mixing ratio and temperature as measured by the Raman lidar are now considered in the study to obtain independent measurements of relative humidity and its time evolution, with a specific focus on the vertical profiles in time coincidence with convective updrafts and downdrafts. Profiles of relative humidity from BASIL have been included in the revised version of figure 6 (formerly figure 5), together with the simultaneous measurements of the particle backscattering coefficient at 1064 nm, $\beta_{1064}$, and particle depolarization at 532 nm, $\delta_{532}$, (from BASIL), and wind direction measurements (from the wind lidar).

… nor the set of particle optical properties is exploited to obtain some microphysical parameters of the aerosol as constraints for the model runs.

In the revised version of the paper some of the particle optical properties measured by BASIL are used to obtain information on aerosol microphysical parameters. Specifically, the comparison of measured and simulated values of the particle backscattering coefficient at 1064 nm allows obtaining estimates of particle number density. We have estimated a particle number density of 0.8-$1.2 \times 10^5$ m$^{-3}$ and $2.5$-$3.5 \times 10^3$ m$^{-3}$ in the small (centered on 4 μm) and large (centered on 20 μm) particle domain, respectively, in agreement with the values reported by various authors (among others, Mészáros, 1991, $0.8$-$3.5 \times 10^5$ m$^{-3}$ for a particle radius of 4 μm and $1$-$2 \times 10^3$ m$^{-3}$ for a particle radius of 20 μm). Additionally, the determination of the backscatter color ratio, *BCR*, (specifically the ratio of total backscattering coefficients at 1064 and 532 nm) and its vertical variability (values found to decrease from 0.40-0.45 below the dark band to 0.33-0.36 within the dark band region) allowed to get an additional confirmation of the conjectured particles' growth during ascent (see more comments and the new introduced sentences below). Furthermore, the variability of the backscatter color ratio with particle radius has also been simulated and the comparison between measured and simulated values of this quantity indicates a particle size in the range 7-11 μm. Finally, a rough estimate of particle sizes is inferred by comparing measured and literature values of both color ratio and depolarization, as in fact values of *BCR* in the range 0.35-0.54 and $\delta_{532}<0.05$ were also reported by Franke *et al.* (2003) and Müller *et al.* (2007) for Southeast Asian aerosols. These aerosols had been found to possess a pronounced coarse mode, being originated mainly from coal and dried plants used for domestic heating and cooking (Müller *et al.*, 2007). Specifically, the determination of particle number concentration and radius allows to impose a constraint in our model runs. It is to be finally pointed out that despite these new results, which allow to constraint our model runs, a number of new citations of additional literature in support of our observations have been introduced (among others, Burton et al., 2012; Burton et al., 2013; Burton et al., 2014; Burton et al., 2015; Dieudonné et al., 2017; Freudenthaler, 2016; Groß, S, 2015; Mészáros, 1991;

Mona et al., 2012; Franke et al., 2003; Müller et al., 2007; Petzold, 2011; Martin, 1993; Mishchenko and Lacis, 2003; Couvreux et al., 2005, 2007; Wulfmeyer et al., 2010, 2016; Turner et al., 2014).

The following new sentences have been introduced in the text: "An additional quantity, namely the backscatter color ratio, $BCR$, i.e. the ratio of total backscattering coefficients at 1064 and 532 nm, was determined from BASIL measurements. Color ratio profiles measured during the time interval considered in the present study (12:00-13:00 UTC on 18 April 2013, not shown here) indicate values in the range 0.40-0.45 below the dark band and in the range 0.33-0.36 within the dark band region. The color ratio decrease is an indication of the increase of particle size. This represents an additional experimental evidence of the conjectured particles' growth, which represents the basis of the given interpretation of the observed phenomenon. Furthermore, small backscatter color ratio values, as those found both below and within the dark band, are indicating relatively large particles (Burton *et al*., 2013), compatible with those conjectured in the present study and presently considered in our simulations. The variability of backscatter color ratio as a function of particle radius has been simulated with the same Mie scattering code already used above, with simulations revealing that values of $BCR$ in the range 0.33-0.45 are compatible with particle size in the range 7-11 µm. Finally, backscatter color ratio values in the range 0.33-0.45 combined with values of $\delta_{532}$ in the range 0.02-0.07 are in agreement with previously observed values of these quantities, as reported by a variety of authors (de Villiers *et al*., 2010, $BCR$=0.3-0.5 and $\delta_{532}$=0.02-0.08; Burton *et al*.; 2014, $BCR$=0.55 and $\delta_{532}$=0.07; Burton *et al*., 2015, $BCR$=0.47 and $\delta_{532}$=0.06-0.09). Similar values ($BCR$=0.35-0.54 and $\delta_{532}$<0.05) were also reported by by Franke *et al*. (2003) and Müller *et al*. (2007) for Southeast Asian aerosols, which were argued to possess a pronounced coarse mode, with large particles being originated mainly from coal and dried plants used for domestic heating and cooking cooking (Müller *et al*., 2007).

The comparison of simulated values of single-particle backscattering coefficient $Q_{back} \times r^2$ ($\sim 3 \times 10^{-11}$ m$^2$ sr$^{-1}$ for a particle radius of 4 µm and $\sim 1 \times 10^{-9}$ m$^2$ sr$^{-1}$ for a particle radius of 20 µm) with measured values of the volume backscattering coefficient $\beta_{1064}$ (in figures 5, in the range 2.5-3.5 \times 10^{-6} m$^{-1}$ sr$^{-1}$ within the dark band) leads to an estimate of particle number density $n$ of 0.8-1.2 \times 10^5 m$^{-3}$ and 2.5-3.5 \times 10^3 m$^{-3}$ in the small and large particles' domain, respectively. These values of $n$ are in agreement with literature values for continental and urban polluted aerosols (among others, Mészáros, 1991, 0.8-3.5 \times 10^5 m$^{-3}$ for a particle radius of 4 µm and 1-2 \times 10^3 m$^{-3}$ for a particle radius of 20 µm)."

And why is only RCS1064 given and not the particle backscatter coefficient (PBC1064), the physically meaningful quantity?

In the revised version of the manuscript the particle backscattering coefficient is considered in substitution to the range corrected signal at 1064 nm ($\beta_{1064}$).

The reviewer understands that the measurement conditions were certainly difficult around noon, so probably water vapor and temperature may not be readily available.

As already mentioned above, vertical profiles of relative humidity, obtained from the simultaneous Raman lidar measurements of water vapor mixing ratio and temperature profiles from BASIL, have now been introduced in the revised version of figure 6 (formerly figure 5). It is to be pointed out that water vapour mixing ratio and temperature profile measurements by BASIL are based on the application of the vibrational and rotational Raman lidar technique, respectively. Both techniques rely on inelastic (Raman) backscatter phenomena, which are characterized by cross-sections that are several orders of magnitude smaller than the cross-sections characterizing the elastic backscatter phenomena. This makes water vapour mixing ratio and temperature measurements, and consequently RH measurements, very difficult to perform, especially in daytime conditions around

noon, as is the case of the measurements illustrated in this paper. This is due to the large solar irradiance affecting the Raman lidar measurements in this portion of the day. This translates into a large statistical uncertainty affecting RH measurements, and consequently large error bars in figure 6, with the random error typically ranging between 4 and 8 % in the altitude region around 1200 m where the particle backscatter minima are observed.

RH profiles measured by BASIL show very similar values in this altitude region during updraft and downdrafts. However, the presence of large statistical uncertainties, and the consequent lack of sensitivity in RH measurements, prevents from revealing any small RH variation between updraft and downdrafts, and consequently prevents from drawing any final conclusion on the motivation behind the absence of the dark-band phenomenon during down-drafts (revealing a small RH variation between updraft and downdrafts would have allowed to justify the absence during particles' descent of a particle growth reversal process, expected from the evaporation of the previously up-taken water, and would have consequently supported our interpretation of the absence of the dark-band feature during down-drafts).

But some sort of microphysical analysis and certainly calculation of PBC1064 should be possible and should be part of this investigation.

In the revised version of the manuscript the particle backscattering coefficient is considered in substitution to the range corrected signal at 1064 nm ($\beta_{1064}$). Additionally, as already mentioned above, information on specific aerosol microphysical parameters is inferred from the multi-wavelength particle backscatter and depolarization measurements. More specifically, an assessment of particle number density in both the small and large range domain is now obtained by comparing measured values of $\beta_{1064}$ with simulated values of $Q_{back}*r^2$ (the latter being the new simulated quantity we are now focusing our attention following the reviewer's suggestion). This comparison allowed to obtained estimates of particle number density: we estimated a particle number density of $0.8\text{-}1.2\mathrm{x}10^5$ m$^{-3}$ and $2.5\text{-}3.5\mathrm{x}10^3$ m$^{-3}$ in the small (centered on 4 μm) and large (centered on 20 μm) particle domain, respectively. These estimated values are found to be in agreement with those reported by various authors (among others, Mészáros, 1991, $0.8\text{-}3.5\mathrm{x}10^5$ m$^{-3}$ for a particle radius of 4 μm and $1\text{-}2\mathrm{x}10^3$ m$^{-3}$ for a particle radius of 20 μm). Additionally, we also determined the color ratio, i.e. the ratio of the total backscattering coefficients at 1064 and 532 nm ($\beta_{1064}/\beta_{532}$). This quantity is known to be dependent of particle size, with a tendency to increase with decreasing particle size. The color ratio is found to have values in the range 0.40-0.45 below the dark band and in the range 0.33-0.36 within the dark band region., i.e. within the same region where the $\beta_{1064}$ reduction of ~ 10 % is observed. This decrease in color ratio indicates an increase of particle size during its uplift. Thus, an additional experimental evidence of the particles' growth conjectured in the interpretation of the observed phenomenon is provided. The text has been modified and integrated with the introduction of the following sentences/paragraphs.
"An additional quantity, namely the backscatter color ratio, *BCR*, i.e. the ratio of total backscattering coefficients at 1064 and 532 nm, was determined from BASIL measurements. Color ratio profiles measured during the time interval considered in the present study (12:00-13:00 UTC on 18 April 2013, not shown here) indicate values in the range 0.40-0.45 below the dark band and in the range 0.33-0.36 within the dark band region. The color ratio decrease is an indication of the increase of particle size. This represents an additional experimental evidence of the conjectured particles' growth, which represents the basis of the given interpretation of the observed phenomenon. Furthermore, small backscatter color ratio values, as those found both below and within the dark band, are indicating relatively large particles (Burton *et al*., 2013), compatible with those conjectured in the present study and presently considered in our simulations. The variability of backscatter color ratio as a function of particle radius has been simulated with the same Mie scattering code already used above, with simulations revealing that values of *BCR* in the range 0.33-0.45 are compatible with particle size in the range 7-11 μm. Finally, backscatter color ratio values

in the range 0.33-0.45 combined with values of $\delta_{532}$ in the range 0.02-0.07 are in agreement with previously observed values of these quantities as reported by a variety of authors (de Villiers *et al*., 2010, *BCR*=0.3-0.5 and $\delta_{532}$=0.02-0.08; Burton *et al*.; 2014, *BCR*=0.55 and $\delta_{532}$=0.07; Burton *et al*., 2015, *BCR*=0.47 and $\delta_{532}$=0.06-0.09). Similar values (*BCR*=0.35-0.54 and $\delta_{532}$<0.05) were also reported by Franke *et al*. (2003) and Müller *et al*. (2007) for Southeast Asian aerosols, which were argued to possess a pronounced coarse mode, with large particles being originated mainly from coal and dried plants used for domestic heating and cooking (Müller *et al*., 2007).

The comparison of simulated values of single-particle backscattering coefficient $Q_{back}{\times}r^2$ ($\sim 3{\times}10^{-11}$ m$^2$ sr$^{-1}$ for a particle radius of 4 µm and $\sim 1{\times}10^{-9}$ m$^2$ sr$^{-1}$ for a particle radius of 20 µm) with measured values of the volume backscattering coefficient $\beta_{1064}$ (in figures 5, in the range 2.5-3.5$\times10^{-6}$ m$^{-1}$ sr$^{-1}$ within the dark band) leads to an estimate of particle number density $n$ of 0.8-1.2$\times10^5$ m$^{-3}$ and 2.5-3.5$\times10^3$ m$^{-3}$ in the small and large particles' domain, respectively. These values of $n$ are in agreement with literature values for continental and urban polluted aerosols (among others, Mészáros, 1991, 0.8-3.5$\times10^5$ m$^{-3}$ for a particle radius of 4 µm and 1-2$\times10^3$ m$^{-3}$ for a particle radius of 20 µm)."

For the purpose of illustrating the results in a more coherent way, former figure 6 (now figure 5), illustrating the time–height cross-section of the particle depolarization ratio at 532 nm, $\delta_{532}$, is now preceding former figure 5 (now figure 6). The text associated with the illustration of new figure 5 and the description of the time–height evolution of $\delta_{532}$ has been slightly modified to account for the refined calculation and calibration of $\delta_{532}$ measurements. In fact, as a results of the more quantitative assessment of the lidar observations requested by the referee, we refined the calculation and calibration of $\delta_{532}$ measurements, obtaining slightly smaller values for this quantity. Now, $\delta_{532}$ ranges from values of 0.05-0.07 below the dark band to values of 0.02-0.03 within and above the dark band. In section 3, after the sentence ending with "respectively (Di Girolamo *et al*., 2012a)", the following sentences have been introduced: "Figure 5 reveals a decrease in particle depolarization at the same height and time intervals of the dark band. More specifically, $\delta_{532}$ decreases from values of 0.05-0.07 below the dark band to values of 0.02-0.03 within and above the dark band. A decrease of $\delta_{532}$ within and above the dark band is compatible with the conjectured size growth of the uplifted dry lignite particles, initially having a more irregular shape, and then getting a more regular spherical shape as a result of the water uptake. Additionally, as previously observed for $\beta_{1064}$, the decrease of $\delta_{532}$ occurs during up-drafts, but not during down-drafts, as in fact during these latter values of $\delta_{532}$ are in the range 0.02-0.04 both below and within the dark band. However, both below and within the dark band values of $\delta_{532}$ are rather low, which is typical of aerosols including a large portion of carbonaceous species as those resulting from fossil fuel combustion, having a rather spherical shape (Dieudonné *et al*., 2017; Müller *et al*., 2007). Particle depolarization ratio measurements, while providing some information on particle shape, may also be used for aerosol typing and mass concentration studies (among others, Petzold, 2011; Burton *et al*., 2012)."

2. Because DR532 is significant, application of Mie theory is questionable. Please, discuss.

As requested by the referee, we performed a more quantitative assessment of the lidar observations trying to go beyond the more qualitative one originally provided in the first version of their paper. As a result of this integration of analysis, we have now refined the calculation and calibration of DR532 measurements ($\delta_{532}$ in the text), obtaining slightly smaller values for this quantity. Now, $\delta_{532}$ ranges from values of 0.05-0.07 below the dark band to values of 0.02-0.03 within and above the dark band, while before the refinement of the calculation and calibration $\delta_{532}$ was ranging from values of 0.30-0.35 below the dark band to values of 0.15-0.20 within and above the dark band. We believe that when dealing with these smaller values of $\delta_{532}$ the application of the Mie theory in the interpretation of the results is less questionable. Nevertheless, the applicability of the Mie theory in

the presence of slightly non-spherical particles is now discussed in the paper. More specifically, we are now clearly pointing out that Mie theory can still be applied for particles characterized by a limited degree of asphericity (Martin, 1993; Mishchenko and Lacis, 2003). In this regard, the following sentence has been introduced in section 4, when illustrating the simulations and their results: "In this respect it is to be specified that the small values of $\delta_{532}$ characterizing the observed aerosol particles call for a very limited degree of asphericity, which makes Mie theory still successfully applicable for the simulation of particles' scattering properties (Martin, 1993; Mishchenko and Lacis, 2003)."

3. Discussion on Figs. 7 and 8, page 6: Why is backscattering efficiency in arbitrary units, and why is backscattering efficiency presented at all?

The referee is write. There was a misprint as in fact the backscattering efficiency was supposed to be expressed in sr$^{-1}$. However, backscattering efficiency ($Q_{back}$) is no longer illustrated in figure 7, having been replaced from the quantity $Q_{back}*r^2$, as suggested by the referee. This is now the quantity compared to measured values of $\beta_{1064}$.

Assuming a constant particle number density n, particle backscatter coefficient is proportional to the product of backscattering efficiency Q and the square of the particle radius, r^2. So, Qr^2 should be modeled to explain the lidar dark band, and ideally it should be compared to the BASIL measurements of PBC1064.

As suggested by the referee, we are now modeling the quantity $Q_{back}*r^2$, which is the quantity now visualized in figure 7. We are now using this quantity also for the interpretation of the observed lidar dark band phenomenon and comparing this quantity to BASIL measurements of $\beta_{1064}$. In this regard, the corresponding text in section 4 of the paper has been changed as follows: "Simulations of the scattering properties of lignite particles are illustrated in figure 7. The figure shows the variability of the quantity $Q_{back}*r^2$ as a function of $r$, with $Q_{back}$, being the backscattering efficiency and $r$ being the particle radius." …. "The quantity $Q_{back}*r^2$ represents the single-particle backscattering coefficient, assuming a constant particle number density $n$." … "The comparison of simulated values of single-particle backscattering coefficient $Q_{back}*r^2$ ($\sim 3\times10^{-11}$ m$^2$ sr$^{-1}$ for a particle radius of 4 μm and $\sim 1\times10^{-9}$ m$^2$ sr$^{-1}$ for a particle radius of 20 μm) with measured values of the volume backscattering coefficient $\beta_{1064}$ (in figures 5, in the range 2.5-3.5x10$^{-6}$ m$^{-1}$ sr$^{-1}$ within the dark band) leads to an estimate of particle number density $n$ of 0.8-1.2x10$^5$ m$^{-3}$ and 2.5-3.5x10$^3$ m$^{-3}$ in the small and large particles' domain, respectively. These values of $n$ are in agreement with literature values for continental and urban polluted aerosols (among others, Mészáros, 1991, 0.8-3.5x10$^5$ m$^{-3}$ for a particle radius of 4 μm and 1-2x10$^3$ m$^{-3}$ for a particle radius of 20 μm)."

Note that a decrease in Q of 10% (top panel) and 40% (bottom panel) as modeled maximally for small and large initial lignite particles (Fig. 7) would be compensated for by an increase in r by a factor of 1.05 and 1.19, respectively.

Authors are not sure to understand the point made by the referee here as in fact it is this "compensation" of the decrease in $Q_{back}$ with an increase of the particle radius (i.e. particle growth process) that we are referring to with the purpose of interpreting the observed phenomena. In the previous version of the paper, former figure 7 was showing a decrease in $Q_{back}$ by $\sim$ 15 % for small initial lignite particles (top panel) and by $\sim$ 40% for large initial lignite particles (bottom panel), which could be caused by an increase of particles' radius $r$ by $\sim$ 15 and 30 %, respectively. More specifically, $Q_{back}$ had been modeled (upper portion of the former version of figure 7) to be equal to 1,565 for $r$ = 2,62 μm and equal to 1,364 for $r$ = 2,99 μm, which corresponds to a decrease in $Q_{back}$ of 13 % associated with an increase of the radius $r$ by 14 %; analogously, $Q_{back}$ had been modeled

(lower portion of the former version of figure 7) to be equal to 4,2 for $r$ = 16,3 μm and equal to 2,3 for $r$ = 21,4 μm, which corresponds to a decrease in $Q_{back}$ of 45 % associated with an increase of the radius $r$ by 31 %; furthermore, $Q_{back}$ was found (former version of figure 7, lower portion) to be equal to 6,1 for $r$ = 27,8 μm and equal to 3,9 for $r$ = 33,0 μm, which corresponds to a decrease in $Q_{back}$ of 36 % for an increase of the radius $r$ by 19 %. This result had been included in the former version of the paper with the sentence: "As a result of these oscillations, for specific radius values of the dry lignite particles (for example, 6.5 μm, 7.5 μm, 18 μm, 28.5 μm, 41 μm), a reduction in $Q_{back} \times r^2$ of 8-27 % (0.35-1.4 dB) is observed for a particle size growth by 10-16 %, which is the size growth experienced by these particles during their adiabatic ascent".

The former considerations of the variability of the quantity $Q_{back}$ as a function of the particle radius $r$ are now confirmed and substantiated by the consideration of the quantity $Q_{back} \ast r^2$, which is the quantity now included in figure 7. Specifically, $Q_{back} \ast r^2$ is found to decrease (upper portion of new figure 7) from an initial value of $9.5 \times 10^{-11}$ m$^2$ sr$^{-1}$ for $r$ = 7.5 μm to a value of $8.8 \times 10^{-11}$ m$^2$ sr$^{-1}$ for $r$ = 8,4 μm, which corresponds to a decrease in $Q_{back}$ of 8 % for an increase of the radius $r$ by 12 %; analogously, $Q_{back} \ast r^2$ is found to decrease (lower portion of the new version of figure 7) from an initial value of $5.2 \times 10^{-11}$ m$^2$ sr$^{-1}$ for $r$ = 28,5 μm to a value of $3.8 \times 10^{-11}$ m$^2$ sr$^{-1}$ for $r$ = 33,0 μm, which corresponds to a decrease in $Q_{back}$ of 27 % for an increase of the radius $r$ by 16 %; finally, $Q_{back} \ast r^2$ is found to decrease (lower portion of the new version of figure 7) from an initial value of $1.28 \times 10^{-8}$ m$^2$ sr$^{-1}$ for $r$ = 41 μm to a value of $1.05 \times 10^{-8}$ m$^2$ sr$^{-1}$ for $r$ = 45 μm, which corresponds to a decrease in $Q_{back}$ of 28 % for an increase of the radius $r$ by 10 %.

4. According to Figs. 3 and 4, westerly winds (> 240 degrees wind direction) prevailed throughout the boundary layer which did not pass over the open-pit mine (Fig. 1). The only exception is the dark-band layer characterized by slightly more southerly winds blowing from the edge of the pit. So isn't it possible that BASIL simply observed air masses of different origin (and therefore different aerosols) at different heights?

The point is that the dark-band is observed during updrafts and not during downdrafts, bearing in mind the shown evident correlation between the presence of the backscatter minimum and the strong positive vertical wind speeds (updrafts) and between the absence of the backscattering minimum and the strong negative vertical wind speeds (downdrafts). This correlation is even more clear in the new version of figures 2, 4 and 5, that we generated following the suggestions of the referee. In this direction, the modified version of figure 6 (formerly figure 5) is very emblematic, as in fact no evidence of a wind direction change is observed when passing from periods with to periods without the dark band. Additionally, again following the requests of the referee, the layout of figure 4 has been improved with the introduction of colored curves and the use of a narrower direction range (180-320 degrees). This new version of figure 4 now clearly reveals that the average wind direction over the time interval 12:00-13:00 UTC on 18 April 2013, i.e. the time interval we are focusing our attention in this paper, has values in the range 230-240 degrees throughout the vertical interval 800-1350 m, which testify the presence of winds blowing from the open-pit mine. A 1h average wind direction profile with values in the range 230-240 degrees reveals that indeed this wind direction was not sporadically experienced during the observation period.
Additionally, what appears quite anomalous here - and calls for a non-dynamical interpretation of the observed phenomenon - is the fact that the region of reduced backscattering persists at an almost fixed height albeit the evident presence of up-drafts and down-drafts, which should have at least perturbed its shape. So, instead of having a time-height cross-section of $\beta_{1064}$ with the dark band appearing as a straight horizontal line feature, in case of a prevailing dynamical cause an alternating structure should be present. Finally, the presence of alternating intensity fluctuations, with the backscatter minimum occurring during up-drafts, but not during down-drafts, would be difficult to explain when considering different aerosol types at different altitudes.

Or asked differently, how certain can one be that really aerosol growth was observed? Please, discuss.

We believe that here the observation of an aerosol growth process is quite likely, but it is obviously not certain. This aspect has been more clearly addressed in the text of the paper with the consideration of a more prudential interpretation of the results and with the introduction in the section "Summary and final remarks" of the following sentence: "Observations and results illustrated in this paper support the interpretation of the phenomenon as a purely microphysical growth mechanism; however, the possibility that other mechanisms (for example, dynamics) may also participate and contribute to the appearance of the phenomenon cannot be completely excluded."

5. Page 7, lines 29ff., page 8, lines 27ff.: In their discussion of the differing aerosol behavior in up- and downdrafts the authors mention the possibility that different particles were measured. The reviewer agrees, see 4. (above). To investigate this important issue, the authors should not only show vertical wind speed for up- and downdraft periods in Fig. 5 but also wind direction.

As suggested by the referee, we integrated the investigation of this issue with the introduction in the modified version of figure 6 (former figure 5) of the wind direction measurements for updrafts (panels a, c, e, g) and downdraft periods (panels b, d, f, h). As additionally suggested by the referee, the vertical wind speed (from the wind lidar) is now illustrated together with the simultaneous measurements of the particle backscattering coefficient at 1064 nm, $\beta_{1064}$, and particle depolarization at 532 nm, $\delta_{532}$, (from BASIL) and vertical wind speed measurements (from the wind lidar). Wind direction profiles in this figure very emblematically reveal that no evidence of a wind direction change is observed when passing from periods with the dark band to periods without the dark band.

In our discussion of the results the possibility that "down-drafts transport other or modified particles than the up-drafts" is considered. However, in this discussion we are primarily referring to the possibility of having entrainment of air from the free troposphere. This is now more clearly specified in the paper, where the text has been changed as follows: "The possibility that particles within the down-drafts are different from those within the up-drafts increases in the interfacial layer due to the entrainment effects and is possibly testified by the presence of smaller particle backscatter values within the down-drafts with respect to those observed within the up-drafts (see figure 2). This is possibly associated with the entrainment of air from the free-troposphere at the top of the CBL, which may ultimately lead to changes in particle size distribution and scattering properties. Evidence of the sharp entrainment of air pockets from the free troposphere into the boundary layer, which gradually mix with the environmental air, has been reported by a variety of authors (Couvreux et al., 2005, 2007; Wulfmeyer et al., 2010, 2016; Turner et al., 2014). Particle size distribution within the down-draft could be not as narrow as in the updrafts, resulting in a smear out of backscatter efficiency oscillations."

Specific comments:

1. Page 3, lines 29, 30: Please, discard RCS1064 in Fig. 3 and show the full profile of PBC1064 instead for a better understanding of the measurement situation.

The particle backscattering coefficient at 1064 nm is now plotted instead of the original range-corrected signal. The full profile of this quantity is now included in the plot.

2. Page 4, line 25: This is no true. Depolarization ratio also depends on size and refractive index of the aerosol particles. Please, be more accurate.

The reviewer is right in underlining that particle depolarization depends not only on the degree of asphericity of sounded aerosol particles, but also on their size and refractive index. The sentence here has been reformulated in order to make it more accurate and now reads: "Particle depolarization ratio, defined as the power ratio of the cross-polarized to the co-polarized components of the particle backscattering coefficient, provides an indication of the degree of asphericity of sounded particles". Additionally, the following new sentence has been introduced: "Particle depolarization depends not only on particles' shape, but also on their size and refractive index (among other, Burton *et al.*, 2015)."

3. Page 5, line 1: The authors probably mean 'Figure 6'. The observation made is not at all obvious. Please, show some example profile pairs (PBC1064, DR532).

Indeed in the text we are referring to figure 6 and not to figure 7. In the new version of this figure (now figure 5) we are now considering a clearer color scale for DR532 in order to make the variability of this quantity easier to reveal and the observations reported in the text more obvious. Besides that, the vertical profiles of PBC1064 ($\beta_{1064}$) and DR532 ($\delta_{532}$) have now been introduced in the new version of figure 6 (formerly figure 5) in order to underline the anti-correlated behavior characterizing the vertical variability of these two quantities.

4. Page 6, line 4: How do the authors arrive at the conclusion of 15-30% size growth? And is it volume, mass, diameter growth?

We assume that lignite particles advected by the wind to the lidar site are captured and ingested within the updrafts and downdrafts associated with the intensive convective activity present at the lidar site. As a result of the adiabatic cooling associated with the uplift, air-parcels undergo a sudden RH increase from values in the range 60-62 % (environmental RH values at the base of the dark band) to values in the range 75-80 % (these latter being the values reached within the lifting air-parcel assuming an ideal adiabatic cooling with no air entrainment into the convective plumes or external air ingestion within the lifting air-parcel).
The solution effect is well known to typically dominate hygroscopic particles' growth when the radius is smaller than the critical radius, which results in small solution droplets being in equilibrium with water vapour at RH values less than 100 %. At this stage, small increases in RH determine particles' size growth until equilibrium is newly reached, i.e. if relative humidity increases by a small amount, the solution droplet grows until equilibrium is reached again. The change in radius associated with a certain variation in relative humidity can be quantified based on the application of the Köhler equation, which is dominated by the solution term when RH values are smaller than 100 %. This term depends on the mass and molecular weight of the solute species and the so called van't Hoff factor. Based on literature values of these quantities, an increase in RH from 60-62 to 75-80 % is expected to determine particles' size growth in radius by 10-20 %. Such percentage increase in particle radius is compatible with the simulated particle radius growth (10-16 %) capable to determine the observed percentage reduction in the backscattering coefficient (~10 %). This aspects are now better clarified in the text. Specifically, when describing the relative humidity change experienced by the aerosol particles during their uplift, the following sentences have been introduced: "At this stage, small increases in RH determine particles' size growth until equilibrium is newly reached. This mechanism is possibly responsible for the lignite particle growth below the LCL, ultimately leading to the appearance of a minimum in lidar backscatter echoes (i.e. the above mentioned clear-air dark band phenomenon). The increase in particles' radius associated with the relative humidity change experienced by the adiabatically uplifted air-parcel can be estimated based on the application of the Köhler equation. When RH values are smaller than 100 %, the Köhler equation is dominated by the solution term, which depends on the mass and molecular

weight of the solute species and the so called van't Hoff factor. Based on literature values of these quantities, the above specified increase of RH from 60-62 to 75-80 % would result in a particle size growth in radius by 10-20 %."

Figures:

All figures: Use the same style, make sure axis labels and titles are easy to read.

We have improved the layout of all figures in the direction suggested by the referee. Now the same style is used in all figures. We also modified the axis labels and titles of several figures in order to make them easier to read.

Figure 2, upper panel: Show PBC1064. Dotted line hard to see. Color bar?

The dotted line in the upper portion of figure 2 has been made thicker in order to make it easier to see. Here the particle backscattering coefficient at 1064 nm is now plotted instead of the original range-corrected signal. The color bar was missing in the upper portion of figure 2 and has now been introduced.

Figure 2, lower panel: Up-/down drafts hardly visible, use different color table.

As suggested by the referee, the color scale has been changed in order to make updrafts and downdrafts easier to see. In this direction, a smaller vertical velocity range has been considered (formerly ± 5, now ± 3), as well as a larger number of colors in the color table.

Figure 3: Show PBC1064, full profile.

The full profile of the particle backscattering coefficient at 1064 nm is now illustrated in figure 3.

Figure 4: Use colored curves instead of symbols, explain curves in caption. Choose narrower direction range so that relevant values can be better judged, for instance 180-320 degrees.

Figure 4 has been changed in the direction suggested by the reviewer. Colored curves are now used instead of symbols. A narrower direction range has been considered in order to make relevant values easier to judge (we are now considering the direction range 180-320 degrees instead of the original 0-360 degrees). All profiles are now explained in the figure caption, which now reads: "Vertical profile of wind speed and direction averaged over the time interval 12:00-13:00 UTC on 18 April 2013 as measured by the wind lidar located in the proximity of BASIL at the Supersite JOYCE. Profiles are reported with error bars, corresponding to ± 1 standard deviation." Finally, the figure was also simplified with the removal of two profiles, representing the minimum and maximum wind speed and direction, which were probably not necessary and overloading the figure.

Figure 5: Good start. Please, show PBC1064 instead of RCS1064. Also include DR532 and wind direction for a better characterization of the particles and the measurement conditions, respectively.

Figure 5 (now renamed figure 6) has been changed and now includes PBC1064 ($\beta_{1064}$) instead of RCS1064. Additionally, in this figure we are now including the vertical profile of particle depolarization at 532 nm ($\delta_{532}$) and wind direction. The inclusion of $\delta_{532}$ in figure 5 imposed some reshuffling of the text. In fact, in the original version of the paper the quantity $\delta_{532}$ had been introduced in figure 6, which was representing the time–height cross-section of $\delta_{532}$ over the same

time interval considered in figure 2 (i.e. 12:00-13:00 UTC on 18 April 2013). The consideration of $\delta_{532}$ in former figure 5 (now figure 6) imposes that former figure 6 (now figure 5) is moved ahead.

Figure 6: As previously noted, it is hard to discern (anti)correlations comparing Figs. 2 and 6 by eye. Example profile pairs (PBC1064, DR532) would help.

In figure 2 and 5 (formerly 6) we are now considering a clearer color scale for PBC1064 ($\beta_{1064}$) and DR532 ($\delta_{532}$), respectively, in order to make their variability easier to reveal. Besides that, as suggested by the referee, the vertical profiles of $\beta_{1064}$ and $\delta_{532}$ have been introduced in figure 6 (formerly figure 5) in order to easily reveal the anti-correlations characterizing the vertical variability of these two quantities.

Technical corrections:

1. Throughout text: Subscripts that are not variables must not be italic

Subscripts have been corrected throughout the text.

2. Abstract, 1st line: Remove ')'at line end

Corrected.

3. Page 1, line 17: '200 m'

Corrected.

4. Page 2, line 8: 'for a selected'

Corrected.

5. Page 3, line 25: 'on other days'

Corrected.

6. Page 4, line 8: 'wind to the'

Corrected.

7. Reference Civis… : Remove 'Jan'

Corrected.

8. Reference Krawczy… : Include 'and' between authors.

Corrected.

9. Reference Yau… : Include 'and' between authors, move to end of list

Corrected.

**Anonymous Referee #2**

General Comments

This paper is very well written and deals with a unique lidar aerosol observation. The case is made for subtle growth by condensation of water on a narrow size distribution of aerosol. Under very specific atmospheric conditions the growth can result in a Mie backscatter minimum at a certain altitude. The lidar and radiosonde data are brought together to make a consistent argument for being able to see this occurrence. This paper is appropriate for ACP and can be published with minor corrections.

We are very pleased for the positive words expressed by the referee. We also appreciate the possibility for further improve the manuscript based on his precious suggestions.

Specific comments

Abstract: no comments

Page 2: Line 23: These eleven detected signals allow(s) determining …

Corrected.

Page 3: Line 15: This minimum persist(s) albeit …

Corrected.

Page 5: Line 1: Figure 7 reveals … Shouldn't that be Figure 6?

The referee is right. This misprint has now been corrected in the revised version of the paper. However, figure 6 has been renamed figure 5 in the new version of the paper.

Figure 7: end of caption: 1.064 micrometers not millimeters

Corrected.

**Anonymous Referee #3**

Summary:

This paper presents an interesting case study of multiwavelength lidar measurements (including linear depolarization ratio measurements) of an unusual aerosol event made during the HOPE campaign in 2013. A thin layer of boundary layer aerosol with an apparently distinctly low associated backscatter coefficient was observed giving rise to a "clear-air dark band". The authors discuss the measurements and offer an explanation for the observations. Namely, they hypothesis that the layer was produced by hydrating lignite particles from a local open-pit mine.
In the main, the paper is clear, the measurements appear sound and the offered explanation seems plausible. The paper is suitable for ACP. However, there are a few areas that should be addressed before publication.

We appreciate the possibility for further improve the manuscript based on the referee's precious suggestions.

While preparing my review I have noticed that Anonymous Reviewer #1 has posted their review, which seems thorough. In the interest of efficiency (and the coming holidays) I will frame my review with reference to Reviewer #1s' comments.

General comments:
I can state that I agree with the general comments of Reviewer#1 with the exception of the contention that "No satisfactory explanation is given why the reversal process likely to occur in downdrafts does not produce a similar effect". I find that the discussion offered in the Section 5 of the paper is plausible and "complete enough" in the context of the present work. Perhaps the authors can state that this is a preliminary hypothesis and outline what exact work would be needed to test their offered explanation in a quantitative manner.

The authors agree that the discussion and interpretation of the results given in the Section 5 was plausible and "complete enough" for the purposes of this paper, whose aim was providing evidence of the observed phenomenon and illustrate possible preliminary explanations for its occurrence. However, the invitation from referee # 1 to integrate the presentation and interpretation of the results offered us the possibility to further investigate the observed phenomena and the issue of the absent reversal process in downdrafts. In this regard, the vertical profiles of relative humidity (RH) from BASIL have been included in the revised version of figure 6 (formerly figure 5), together with the simultaneous measurements of the particle backscattering coefficient at 1064 nm, $\beta_{1064}$, and particle depolarization at 532 nm, $\delta_{532}$, (from BASIL) and wind direction and vertical wind speed measurements (from the wind lidar). RH values observed by BASIL in this altitude region are found to be very similar during updraft and downdrafts. This new version of the figure provides additional evidence of the correlated appearance of backscatter minima (dark band) in coincidence of strong updrafts (positive vertical velocity values) and the disappearance of these backscatter minima in coincidence of strong downdrafts (negative vertical velocity values). Concerning the RH measurements, it is to be pointed out that these are affected by large uncertainties. The lack of sensitivity in RH measurements prevents from revealing any small RH difference between updraft and downdrafts, and consequently prevents from drawing any final conclusion on the presence of a RH change, which could justify the absence during particles' descent of a particle growth reversal process, expected for the evaporation of the previously up-taken water, and could consequently motivate the absence during down-drafts of a dark-band feature. Additionally, the following modified text has been introduced in the discussion of the missing reversal process during downdrafts: "The possibility that particles within the down-drafts are different from those within

the up-drafts increases in the interfacial layer due to the entrainment effects and is possibly testified by the presence of smaller particle backscatter values within the down-drafts with respect to those observed within the up-drafts (see figure 2). This is possibly associated with the entrainment of air from the free-troposphere at the top of the CBL, which may ultimately lead to changes in particle size distribution and scattering properties. Evidence of the sharp entrainment of air pockets from the free troposphere into the boundary layer, which gradually mix with the environmental air, has been reported by a variety of authors (Couvreux et al., 2005, 2007; Wulfmeyer et al., 2010, 2016; Turner et al., 2014)."

I can also state that I especially agree with Reviewer#1 that the Figures need to be, in general, improved.

Following the suggestion of both referee # 1 and 3, the layout of all figures has been improved and we are now using the same style for all of them. We also modified the axis labels and titles of several figures in order to make them easier to read.

Specific Additional Comment

-Line 30 on Page 4 refers to Figure 6: I think this should be Figure 5 OR the authors have left a figure out by mistake. The discussion makes me think they are referring to a line-plot of the depolarization ratio vs height. If this is not the case, and they are really mean to be referencing Figure 5, then they should consider inserting a line plot of the depolarization. In any case, Figure 5 is not clear enough for me to draw any quantitative information from !

The sentence in line 30 of Page 4 (*"However, accurate measurements of these quantities (i.e. the cross- and co-polarized components of the particle backscattering coefficient) may be difficult to obtain, often as a result of the depolarizing properties of different optical devices included in the receiver (Freudenthaler, 2017)"*) refers to former figure 6 (now figure 5), which is the figure illustrating the time-height cross-section of particle depolarization at 532 nm. We don't understand why the referee thinks this sentence should refer to former figure 5 (now figure 6), which is instead illustrating the vertical profiles of the particle backscattering coefficient at 1064 nm ($\beta_{1064}$), formerly the range-corrected backscatter signal at 1064 nm ($RCS_{1064}$), and the vertical wind speed. However, a line-plot of the depolarization ratio vs height has now been introduced in former figure 5 (now figure 6), together with the vertical profiles of the particle backscattering coefficient at 1064 nm, relative humidity, wind direction and vertical wind speed. We have the impression that the statement *"Figure 5 is not clear enough for me to draw any quantitative information from"* by the referee is intended to refer to former figure 6 (now figure 5) and not former figure 5 (now figure 6) as in fact former figure 6 figure was a color map illustrating the time-height variability of particle depolarization without a clear color scale and this was making very difficult to properly infer the values of this quantity, while former figure 5 was illustrating the vertical profiles of several quantities (range corrected signal at 1064 nm, vertical wind velocity, now also particle backscattering coefficient at 1064 nm, relative humidity and wind direction) from which quantitative information could be easily drawn. Nevertheless, we sensitively improved the layout of both figures so that now quantitative information can be easily drawn.